# Histoanatomic Features Distinguishing Aganglionosis in Hirschsprung’s Disease: Toward a Diagnostic Algorithm

**DOI:** 10.3390/diseases13080264

**Published:** 2025-08-16

**Authors:** Emma Fransson, Maria Evertsson, Tyra Lundberg, Tebin Hawez, Gustav Andersson, Christina Granéli, Magnus Cinthio, Tobias Erlöv, Pernilla Stenström

**Affiliations:** 1Department of Pediatric Surgery, Lund University, Skåne University Hospital, 22185 Lund, Sweden; tyra.lundberg@skane.se (T.L.); tebin.hawez@med.lu.se (T.H.); christina.graneli@med.lu.se (C.G.); pernilla.stenstrom@med.lu.se (P.S.); 2Department of Clinical Sciences Lund/Biomedical Engineering, Lund University, 22185 Lund, Sweden; maria.evertsson@med.lu.se; 3Department of Clinical Genetics, Pathology and Molecular Diagnostics, Laboratory Medicine, Medical Services, Skåne University Hospital, 22185 Lund, Sweden; gustav.l.andersson@skane.se; 4Department of Biomedical Engineering, The Faculty of Engineering, Lund University, 22185 Lund, Sweden; magnus.cinthio@bme.lth.se (M.C.); tobias.erlov@bme.lth.se (T.E.)

**Keywords:** Hirschsprung’s disease, diagnostics, ultra-high frequency ultrasonography, frozen biopsy, histopathology

## Abstract

Background/Objectives: Intraoperative frozen biopsies are essential during surgery for Hirschsprung’s disease (HD). However, this method has several limitations with the need for a faster and real-time diagnostic alternative. For this, consistent histoanatomical and morphometric differences between aganglionic and ganglionic bowel must be established. The primary objective was to compare dimensions of bowel wall layers between aganglionic and ganglionic segments histopathologically in resected rectosigmoid specimens from children with HD. Secondary objectives were to design a diagnostic algorithm to distinguish aganglionosis from ganglionosis and assess whether full bowel wall thickness correlates with patient weight and age. Methods: Each histoanatomic bowel wall layer—mucosa, submucosa, and muscularis propria’s layers—was delineated manually on histopathological images. Mean thicknesses were calculated automatically using an in-house image analysis software. Paired parametric tests compared measurements in aganglionic and ganglionic segments. Results: Resected specimens from 30 children with HD were included. Compared to aganglionic bowel, ganglionic bowel showed a thicker muscularis interna (mean 0.666 mm versus 0.461 mm, CI −0.257–(−0.153), *p* < 0.001), and a higher muscularis interna/muscularis externa ratio (2.047 mm versus 1.287 mm, CI −0.954–(−0.565), *p* < 0.001). An algorithm based on these features achieved 100% accuracy in distinguishing aganglionosis from ganglionosis. No significant difference in full bowel wall thickness was found between aganglionic and ganglionic segments, nor any correlation with patient weight or age. Conclusions: Histoanatomic layer thickness differs between aganglionic and ganglionic bowel, forming the basis of a diagnostic algorithm. Full bowel wall thickness was independent of patient weight and age.

## 1. Introduction

Hirschsprung’s disease (HD) is a rare congenital disorder characterized by the absence of ganglion cells in the myenteric and submucosal plexuses of the bowel wall [1]. Surgical resection of the aganglionic segment is required, and intraoperative frozen biopsies are essential to confirm the presence of ganglion cells and ensure complete resection [2,3]. However, the frozen biopsy method has several limitations, such as prolonged anesthesia time [4] and risk of suboptimal resection length since the biopsy site is selected based on the surgeon’s visual approximation [5]. These challenges highlight the need for a faster and real-time diagnostic alternative to frozen biopsies. In this context, further exploration of histomorphometric characteristics may offer valuable insights.

Histological differences in bowel wall layer thickness between aganglionic and ganglionic segments have previously been described in fresh ex vivo specimens using manual measurement techniques [6]. However, the precise morphometric differences between aganglionic and ganglionic rectosigmoid bowel in children with HD, as calculated by computational methods, have not been explored. Additionally, it is unclear whether bowel wall thickness differs with the patient’s weight and age.

The primary aim of this study was to investigate whether histoanatomic layer dimensions differ between aganglionic and ganglionic bowel segments in histopathological specimens, using an in-house developed software. Secondary aims were: (1) to assess whether these measurements could be integrated into a diagnostic algorithm to distinguish aganglionic from ganglionic segments, and (2) to evaluate whether bowel wall thickness correlates with patient weight or age. Based on clinical observations and previous manual measurements [6], we hypothesized that histoanatomic morphometric differences would be significant between aganglionic and ganglionic bowel segments, and that bowel wall thickness would be associated with the patient’s weight and age.

## 2. Methods

### 2.1. Patients and Data

This was an observational study involving children who underwent surgery for rectosigmoid HD between April 2018 and January 2024. Patients were eligible if they met the following inclusion criteria: age below 1 year at time of surgery, primary surgery for HD without prior stoma and a maximum aganglionic segment length of 25 cm. The study was conducted at the National Referral Center for HD, serving a population of approximately 5 million residents. Patient characteristics, including gender, weight at birth, and age and weight at surgery were collected from a local HD registry.

### 2.2. Specimen Treatment and Histopathological Analysis

Freshly resected bowel specimens were pinned onto a cork mat, with the aganglionic and ganglionic bowel ends oriented in opposite directions. Histopathological processing was performed at a tertiary National Center for Pediatric Pathology. Resection specimens were fixed in formalin, followed by careful sampling of tissue critical for defining the extent of the aganglionic segment, the transition zone and expectantly confirming ganglionosis in the proximal resection margin. Tissue samples were embedded in paraffin blocks from which sectioned specimens were stained with hematoxylin and eosin for histological evaluation. Additionally, immunohistochemical staining was performed using calretinin and S100 to further characterize the presence of ganglion cells and nerve structures. All stained slides were analyzed by a specialist in pediatric pathology.

### 2.3. Morphometrical Methods: Histological Layers

For each patient, hematoxylin and eosin-stained slides representing aganglionic and ganglionic tissue, respectively, were selected by a pathologist. The scanned histopathological slides, stored routinely in the hospital’s diagnostic image processing application ‘Sectra PACS/IDS7’ (Sectra), were then transferred to MATLAB (MathWorks Inc., Natick, MA, USA). The MATLAB software (Matlab R2024a) was customized by engineers at the university’s Faculty of Engineering. It has been validated for ultra-high frequency ultrasound applications and used to perform histoanatomic measurements for correlation analysis [7,8]. The delineators (clinicians E.F. and T.L.) were blinded to all clinical information during the assessment of histological sections, although information regarding aganglionosis or ganglionosis may have been available. The software was designed to facilitate manual assessment of the following histoanatomic layers: muscularis propria externa (longitudinal muscle, hereafter referred to as the muscularis externa), myenteric tissue layer (located between the muscularis propria externa and muscularis propria interna), muscularis propria interna (circular muscle, hereafter referred to as the muscularis interna), submucosa, mucosa and the full bowel wall and to calculate the thicknesses of each layer semi-automatically. The inner and outer borders of each layer were delineated manually in the software (Figure 1a) including the entire circular cross-section. In cases where artifacts were present, only a portion of the specimen was delineated. For every histoanatomical layer, mean thickness with standard deviations, total area, inner and outer lengths based on the delineated borders were computed automatically from measures with intervals of 14 µm (Figure 1b). Furthermore, a ratio of muscularis interna to externa (muscularis interna/muscularis externa) was calculated from the mean thicknesses, to investigate their interrelationship in each patient. The degree of submucosal folding was quantified as the ratio of the inner and outer submucosal lengths to assess any differences in the extent of folding between aganglionic and ganglionic bowel.

### 2.4. Statistical Analysis

Statistical analyzes were performed in SPSS Statistics (version 30; IBM, Armonk, NY, USA). Normal distribution of data was confirmed through distribution analyzes performed by statisticians at the Department of Clinical Studies, Statistics, Forum South. The power calculation was based on data from a previous study in which manual measurements were performed [6]. Each patient served as their own control. Paired *t*-tests were used to evaluate systematic differences in histoanatomical layer thickness between aganglionic and ganglionic bowel. For correlations between full thickness bowel measurements and patients’ weight and age at surgery, the Pearson’s correlation test was used. A high correlation was considered if the Pearson’s correlation coefficient was 1 or −1 while no correlation was considered if the coefficient was zero. A *p*-value of less than 0.05 was considered to be statistically significant.

### 2.5. Ethical Considerations

Ethical approval for the study was granted by the local review board (DNR 2017/769) and the Swedish Ethical Review Authority (DNR 2023-01833-01). Oral and written information was given and written consent was obtained from the guardians.

## 3. Results

### 3.1. Patients

A total of 49 patients underwent surgery for HD during the study period. Eighteen patients were excluded for not meeting the inclusion criteria, distributed as: age over 1 year at the time of surgery (*n* = 5), stoma prior to surgery (*n* = 6), not primary surgery for HD (*n* = 1), and aganglionosis length > 25 cm (*n* = 6). Additionally, one patient was excluded due to poor histological image quality of the aganglionic specimen. Following these exclusions, 30 patients were included in the study. Descriptive characteristics of the study population are presented in Table 1. Among the included patients, three had trisomy 21, one trisomy X, and one Mowat Wilson syndrome.

### 3.2. Histological Layers of Bowel Wall

Comparison of layer thickness between aganglionic and ganglionic bowel specimens revealed a significantly greater thickness of the muscularis interna in ganglionic tissue (0.666 mm, SD 0.177) compared to aganglionic bowel (0.461 mm, SD 0.130; *p* < 0.001) (Table 2 and Figure 2). A thicker muscularis interna in ganglionosis was observed in 28/30 patients (Figure 3a). The calculated ratio of muscularis interna to muscularis externa was significantly higher in ganglionic bowel (2.047 mm, SD 0.365) compared to aganglionic bowel (1.287 mm, SD 0.356; *p* < 0.001), with 29/30 patients presenting with a greater ratio in ganglionosis (Figure 3b). Importantly, all ganglionic specimens demonstrated a muscularis interna to muscularis externa ratio greater than 1 (30/30). No statistically significant differences were observed when comparing the thickness of the other histoanatomic layers (muscularis externa, myenteric tissue layer, submucosa, mucosa and the full bowel wall). No significant difference in submucosal folding was observed between aganglionic and ganglionic bowel segments (Table 2). In children with a stoma prior to surgery, the ratio of muscularis interna to externa was significantly greater in ganglionic bowel in all 6 patients, and the muscularis interna was thicker in 5 out of these 6 patients (Appendix A, Table A1).

### 3.3. Diagnostic Algorithm

Our findings support the development of a diagnostic algorithm to distinguish aganglionic from ganglionic bowel, with the following criteria (Figure 4):(I).Thinner muscularis interna in aganglionic compared to ganglionic bowel,(II).Lower muscularis interna/muscularis externa ratio in aganglionic bowel.

Using this algorithm, all specimens were classified correctly. Two specimens showed a thicker muscularis interna in the aganglionic bowel, but both still had a lower muscularis interna to muscularis externa ratio in aganglionic segment. One specimen had a higher ratio in the aganglionic segment, but the muscularis interna was still thinner than in the ganglionic bowel.

### 3.4. Correlation to Age and Weight

Full thickness bowel wall did not correlate significantly to the patient’s weight at surgery (Pearson’s correlation coefficient) neither for aganglionosis (r = 0.063, *p* = 0.740) nor ganglionosis (r = 0.252, *p* = 0.180) (Figure 5a). In addition, full thickness bowel wall did not correlate to the patient’s age at surgery in aganglionosis (r = −0.033, *p* = 0.863) nor ganglionosis (r = 0.230, *p* = 0.222) (Figure 5b).

There was no difference in the full bowel thickness comparing aganglionosis and ganglionosis (Table 2).

## 4. Discussion

In this study, histopathological morphometrics in resected rectosigmoid bowel in children with HD were compared. A key novelty lies in the use of the in-house designed MATLAB software that enabled precise, semi-automated measurements. The main findings were that both the muscularis interna and the muscularis interna to muscularis externa ratio were significantly thicker/greater in ganglionic than in aganglionic bowel. These findings formed the basis for a diagnostic algorithm, successfully distinguishing aganglionic from ganglionic specimens in all cases. No significant difference was observed in full bowel wall thickness between aganglionic and ganglionic bowel. Nor did full bowel wall thickness correlate with the patient’s weight or age at surgery.

Compared to the previous study [6], the strength of this study lies in the high-resolution, semi-automated analysis performed in our software, computing multiple measurement points at 14 µm intervals across the circular cross section. This is of importance since manual measurements are limited by subjectivity, are more prone to interobserver variability and are time-consuming. In addition, a larger number of specimens were included.

Regarding the result with a thicker muscularis interna and higher muscularis interna to muscularis externa, there is very scarce literature to relate to. The only other human study on this subject in histological specimens was conducted by our research group, finding similar results with a thicker muscularis interna in ganglionic specimens compared to aganglionic [8]. The consistency strengthens the validity of our findings.

In line with our results, a murine model of genetically induced congenital aganglionosis in the distal colon demonstrated a thicker muscular layer in the ganglionic bowel, particularly in the muscularis interna. The authors suggested this was due to muscle hypertrophy in response to increased resistance from the aganglionic segment [9]. Clinically, such hypertrophy could then be expected to be less pronounced in patients having effective wash outs or stoma establishment proximal of the aganglionosis. Conversely, two studies using rat models, inducing bowel obstruction mimicking aganglionosis, demonstrated an increased mass and thickness of the muscularis propria in the “aganglionic” bowel [10,11]. This hypertrophy was suggested by the authors to be a result of compensation for a greater radial force origin from an increased mucus content in the aganglionic bowel. Unlike children with HD, the rat models were initially healthy, and the artificial obstruction started acutely rather than being developed chronically as in HD. This might hypothetically influence the hypertrophic development. Moreover, one of the studies examined the jejunum rather than rectosigmoid colon [11], making direct comparison difficult due to anatomical and functional differences.

Recent studies suggest that tissue differences across various conditions can be identified using ultra-high frequency ultrasound examinations [12,13]. In the gastrointestinal tract, ultra-high frequency ultrasound has shown potential in distinguishing aganglionic from ganglionic segments by detecting differences in bowel wall architecture [14,15], and has also been used to identify histomorphological changes associated with bowel inflammation in infants [16].

As in our previous study [6], no difference in full bowel wall thickness was observed between aganglionic and ganglionic bowel, but in contrast no correlation with the patient’s weight or age at surgery was indicated. These findings are important, as they suggest that ultra-high frequency ultrasound assessment in infants can be performed using a consistent transducer type and frequency setting, regardless of whether the bowel is affected by aganglionosis, or the child’s weight and age. Supporting these results, one study suggested no significant increase in colonic wall thickness in healthy individuals with age from 1 month to 39 years [17], while another reported a gradual increase by age, peaking at 2.0 mm in individuals aged 20–29 years [18].

The hypothesis that submucosal folding differs between aganglionic and ganglionic bowel arose from a clinical observation made during microscopic examination. However, neither the thickness or the extent of folding in the submucosa showed any significant differences. A possible explanation for the lack of differences is that submucosal thickness and degree of folding might depend largely on interindividual variation in extracellular matrix accumulation which potentially can cause a fibrosis remodeling in aganglionosis [19]. Another explanation could be methodological, the software was coded to calculate the folding as a relationship between the inner and outer length of the submucosa layer, instead of the depth of the folds directly. Future refinements of the software could enable more precise measurements, which may be particularly relevant when examinations—such as those using ultra-high frequency ultrasound—are performed from the mucosal side of the bowel.

A major strength of this study is the high quality of most of the histopathological images and, in cases of partially available bowel specimens, the possibility to delineate that part. Additionally, by using each patient as their own control, interindividual variability effects were minimized. One limitation was that aganglionic and ganglionic bowel specimens were not always obtained from the exact same colonic level. The reason for this was that image quality was prioritized over anatomic site. Also, the manual delineation process was not fully blinded, introducing the potential for observation bias. The main concern is that the observed differences in muscularis thickness may partly reflect natural anatomical variation between the distal rectum and proximal sigmoid colon—differences that may also be present in children without HD. This hypothesis encounters difficulties to prove ethically, as healthy children do not undergo surgery, making histological control specimens extremely difficult to obtain. Another limitation is that not all characteristics of the bowel wall can be captured solely through anatomical measurements of the muscularis propria. For example, in patients with HD, ganglion cells may be histologically present in the ganglionic bowel—leading the algorithm to register a “positive” result—yet the number, distribution, or functional maturity of these ganglion cells may still be abnormal. This represents a potential diagnostic limitation, particularly in cases where HD is not isolated but part of a broader spectrum of intestinal dysganglionosis, such as intestinal neuronal dysplasia or hypoganglionosis. However, a rectal or colonic ultra-high frequency ultrasound probe for infants could enable non-invasive measurements also in healthy children [20].

## 5. Conclusions

The findings of a thicker muscularis interna and a higher ratio of muscularis interna to muscularis externa in ganglionic bowel provide valuable insights into the histoanatomy of HD. The proposed diagnostic algorithm, achieving 100% accuracy in distinguishing aganglionosis from ganglionosis. However, further validation of the algorithm is required to establish reliable cut-off values. The ultimate goal is to translate this knowledge into real-time diagnostic assessment of the bowel wall.

## Figures and Tables

**Figure 1 diseases-13-00264-f001:**
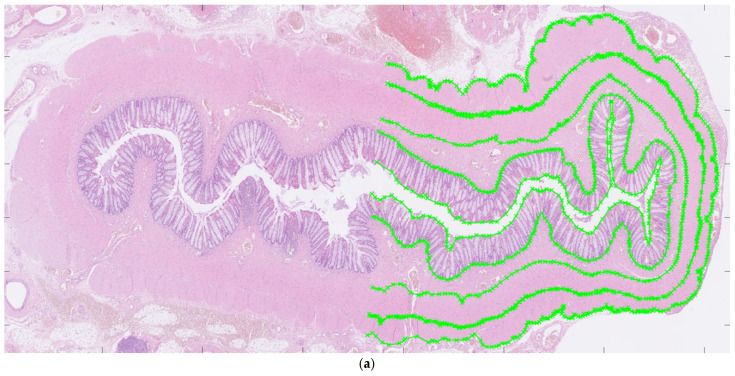
Images showing full circular cross-sections of the aganglionic segment (**a**) and the ganglionic segment (**b**) from a child who underwent surgery for Hirschsprung’s disease. In (**a**), manual delineation of histoanatomical layers performed in the software is illustrated. In (**b**), each white line indicates a measurement point used for the automated calculation of histoanatomical layer thicknesses, also conducted in the software. For illustrative purposes, a half-circumferential delineation was applied in both images.

**Figure 2 diseases-13-00264-f002:**
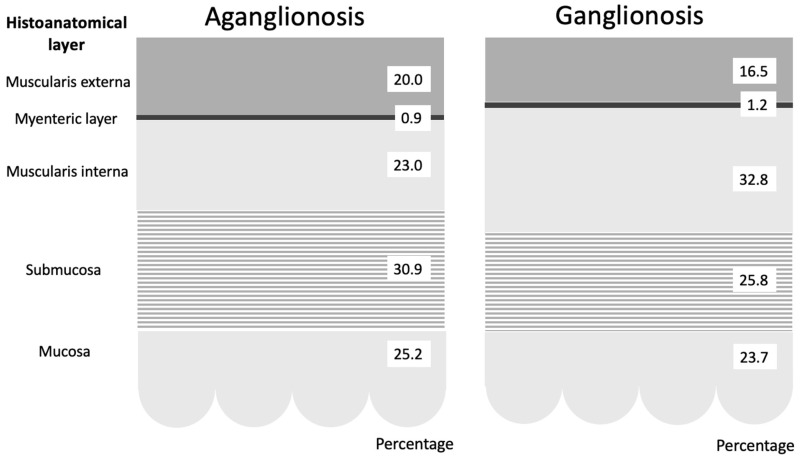
Graphical representation of the average percentage distribution of histoanatomical layers in aganglionic and ganglionic bowel specimens. The figure demonstrates that the most prominent difference was in the muscularis interna, which was significantly thinner in aganglionic compared to ganglionic bowel.

**Figure 3 diseases-13-00264-f003:**
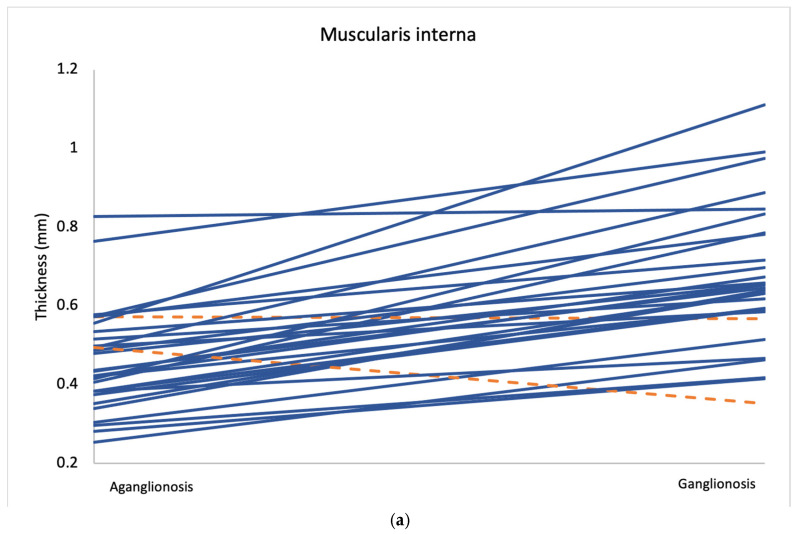
Average measurements of muscularis interna thickness (**a**) and the muscularis interna to muscularis externa ratio (**b**) in aganglionic and ganglionic bowel segments from all 30 patients. Cases where the aganglionic segment showed lower values than the ganglionic segment are marked in blue, while cases with higher values in the aganglionic segment are marked in orange.

**Figure 4 diseases-13-00264-f004:**
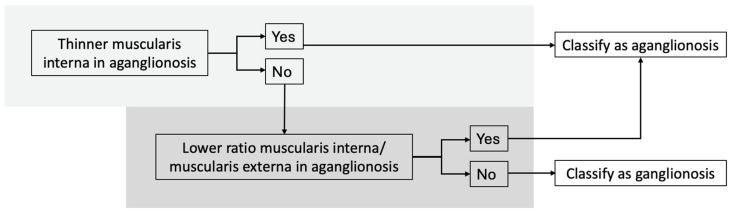
Diagnostic algorithm for distinguishing between aganglionic and ganglionic bowel based on histomorphometric measurements. Using this algorithm, all specimens in our study were classified correctly.

**Figure 5 diseases-13-00264-f005:**
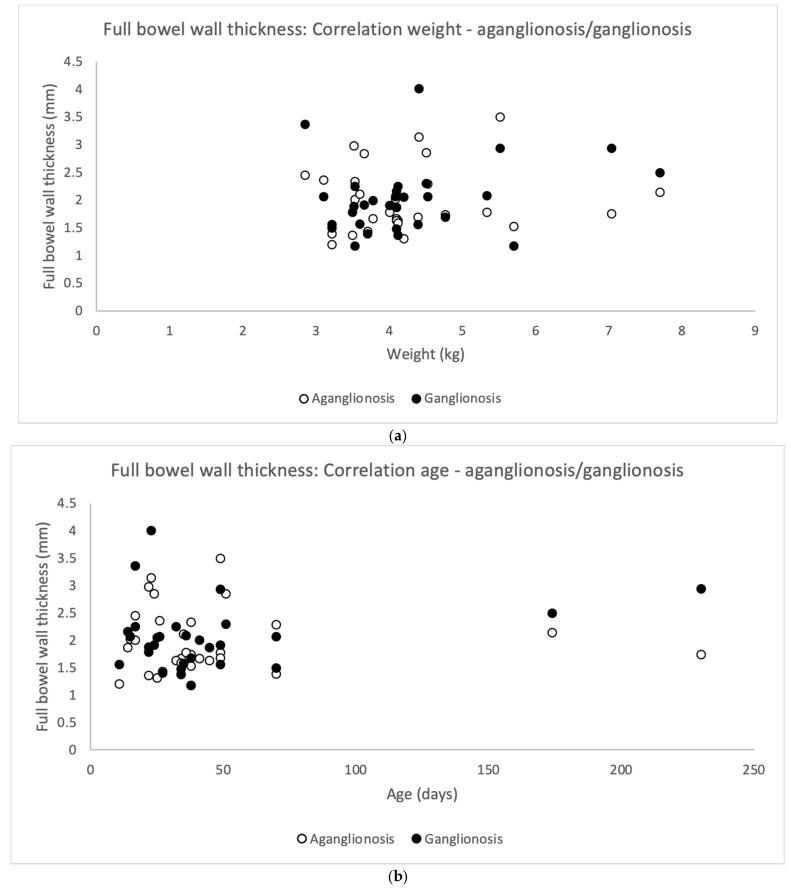
Scatterplots illustrating correlation between full bowel wall thickness in aganglionic and ganglionic specimens and the patient’s (*n* = 30) weight (**a**) and age (**b**) on the day of surgery. No statistically significant correlation was found in aganglionic bowel wall thickness (white dots) correlated to weight (Pearson’s correlation coefficient r = 0.063, *p*-value = 0.740) and to age (r = −0.033, *p*-value = 0.863). Similarly, no statistically significant correlation was observed between ganglionic bowel wall thickness (black dots) and the patient’s weight (r = 0.252, *p*-value = 0.180) or age (r = −0.230, *p*-value = 0.222).

**Table 1 diseases-13-00264-t001:** Descriptive characteristics of the 30 patients with Hirschsprung’s disease included in the analysis of histoanatomic thicknesses in aganglionic versus ganglionic bowel.

	Median	Mean	Interquartile Range
Age at surgery (days)	34.5	45.2	23–49
Weight at surgery (kg)	4.1	4.3	3.5–4.5
Length of aganglionosis (cm)	11.0	11.6	6.0–16.3
Resection length (cm)	18.5	18.0	14.4–21.3

**Table 2 diseases-13-00264-t002:** Systematic differences of histoanatomic measurements in bowel wall comparing aganglionic and ganglionic bowel specimens resected from the 30 children who underwent surgery for Hirschsprung’s disease. Paired samples *t*-test was used as patients served as their own controls. Mean values with standard deviation (SD) presented represent the mean of all individuals’ mean values measured in aganglionic and ganglionic bowel, respectively.

Histoanatomical Layer	Thickness Aganglionosis Mean (SD)	Thickness Ganglionosis Mean (SD)	Difference Mean (SD) Range	Confidence Interval of the Difference	*p*-Value *	Thicker/Greater in Ganglionosis Versus Aganglionosis (n)
Muscularis externa (mm)	0.401 (0.203)	0.334 (0.100)	0.066 (0.204)	−0.010–0.143	0.086	15
Muscularis interna (mm)	0.461 (0.130)	0.666 (0.177)	−0.205 (0.138)	−0.257–(−0.153)	<0.001	28
Ratio m. interna/m. externa	1.287 (0.356)	2.047 (0.365)	−0.760 (0.521)	−0.954–(−0.565)	<0.001	29
Myenteric tissue layer (mm) ^1^	0.018 (0.018)	0.024 (0.017)	−0.006 (0.020)	−0.014–0.001	0.092	21
Submucosa (mm)	0.618 (0.244)	0.524 (0.277)	0.094 (0.280)	−0.011–0.198	0.076	8
Mucosa (mm)	0.505 (0.191)	0.481 (0.159)	0.024 (0.187)	−0.046–0.094	0.482	12
Full bowel (mm)	2.003 (0.586)	2.030 (0.630)	−0.027 (0.569)	−0.239–0.186	0.799	16
Ratio submucosa inner/outer length (folding)	1.101 (0.203)	1.080 (0.132)	0.021 (0.222)	−0.062–0.104	0.611	12

^1^ missing *n* = 1. * Paired samples *t*-test.

## Data Availability

The data presented in this study are available from the corresponding author upon reasonable request.

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
