# Peer review of "Histoanatomic Features Distinguishing Aganglionosis in Hirschsprung’s Disease: Toward a Diagnostic Algorithm"

_diseases, 2025, doi:10.3390/diseases13080264_

Round 1

Reviewer 1 Report

Comments and Suggestions for Authors

This manuscript presents an original and methodologically well-designed contribution to the diagnostic approach in Hirschsprung’s disease. The authors used precise morphometric analysis of histological sections to identify consistent differences in muscularis propria layer thickness between aganglionic and ganglionic bowel. The key finding — significantly thicker muscularis interna and a higher muscularis interna/muscularis externa ratio in ganglionic bowel — provides a solid basis for a diagnostic algorithm that could potentially replace intraoperative frozen biopsies.

This retrospective study included 30 patients, each serving as their own control (paired comparisons). Histological sections were analyzed using a MATLAB-based software specifically developed for this purpose. The statistical methods are appropriate and clearly presented. Of particular note is the high quality of most histological specimens and robust ethical considerations.

Based on this model, the use of UHFUS combined with the histoanatomic algorithm enables precise identification of the transition zone between aganglionic and ganglionic segments, thereby allowing determination of the optimal resection site during surgery for Hirschsprung’s disease. This represents a significant step forward toward non-invasive, real-time diagnostics.

However, it is important to emphasize that Hirschsprung’s disease is not always an isolated entity. In some cases, HD is part of a broader spectrum of intestinal dysganglionoses — such as intestinal neuronal dysplasia and hypoganglionosis. In such cases, although ganglion cells may be histologically present (meaning the algorithm and UHFUS would yield a "positive" result for ganglionosis), their functional maturity, distribution, or number may be pathological — which cannot be identified based solely on the anatomic dimensions of the muscularis propria layers. I suggest that these limitations also be included in the manuscript. Another of the limitations of the study is that the aganglionic and ganglionic tissue samples were not always taken from the same anatomical level, which may introduce variability (included in the manuscript).

It would be beneficial to demonstrate clinical validation of the algorithm in real-time using UHFUS, which would represent the next step toward its implementation in routine clinical practice

This study makes a significant contribution to the histoanatomic understanding of Hirschsprung’s disease and provides a strong foundation for developing faster, objective, and non-invasive diagnostic methods. I recommend the manuscript for publication with only minor revisions.

Accept with minor revision.

Reviewer 2 Report

Comments and Suggestions for Authors

Congratulations to Fransson et al. on presenting an interesting study that investigates histoanatomic differences between aganglionic and ganglionic bowel segments in Hirschsprung’s disease and explores the potential role of ultra-high-frequency ultrasonography as a real-time diagnostic tool.

The topic is clinically relevant and, overall, well-written

The title is lengthy and may confuse readers. 

In the introduction, the manuscript contains a clear description of the disease background and a strong justification for examining layer-specific bowel wall dimensions. The overall objective is described through three separate aims, but the wording is complicated. I suggest condensing this study's goal description.

The methods section explains the eligibility criteria, data sources, and histologic measurements comprehensively. One point remains unclear: was the pathologist performing the measurements blinded to clinical information?

It would also be helpful to report whether any a priori sample size calculation was performed to justify the study's sample size and to ensure that the study is adequately powered for paired comparisons.

Results are presented in a logical manner, and the figures effectively illustrate the key morphometric differences. Nonetheless, the transformation of these findings into a “diagnostic algorithm” appears premature. The manuscript does not establish explicit quantitative thresholds or internal/external validation steps, both of which are necessary before an algorithm can be considered clinically actionable. Presenting the proposed ratio and thickness criteria as hypotheses for future prospective evaluation would be more appropriate than framing them as a validated diagnostic tool.

The conclusion should therefore be tempered, noting that the work demonstrates measurable differences in muscularis interna thickness and its ratio to muscularis externa, but that further studies are required to define reliable cut-offs, test inter-observer reproducibility, and confirm performance in independent cohorts.

Reviewer 3 Report

Comments and Suggestions for Authors

The article provides a new incentive for the precise diagnosis of HD and is a relevant manuscript to the field.

It is pretty much well written, with a rather good description of the methods and statistics, as well as the discussion section. However, perhaps the most confusing aspect, and the authors should clearly explain, because this should be easily acknowledged by all readers, is the role of UHFUS in this article.

The main objective is to use MATLAB software, validated for UHFUS applications, that may perform correlation analysis. The introduction is mainly focused on the UHFUS, which means ultra-high frequency ultrasonography. They describe it’s potential, and how it may impact future surgeries; however, as I advanced through the manuscript, no relevant information is available on ultrasound on those patients. No images, no parameters noted on US imaging. I can understand the software option and potential, but your article does not have any UHFUS measurements, only pathology measurements. Either restructure the entire article and provide specific US imaging in comparison with pathology assessment based on relevant software that may combine ultrasound and pathology parameters, or just mention the technique with future potential.

Reviewer 4 Report

Comments and Suggestions for Authors

Dear Editor,

The authors histopathologically compared the dimensions of the intestinal wall layers between the aganglionic and ganglionic segments in resected rectosigmoid specimens from children with Hirschsprung's disease. They also aimed to design a diagnostic algorithm to differentiate aganglionosis from ganglionosis and to evaluate whether complete bowel wall thickness correlates with patient weight and age. Resected specimens from 30 children with HD were included in the study. Compared with aganglionic bowel, ganglionic bowel had a thicker muscularis interna and a higher muscularis interna/muscularis externa ratio. An algorithm based on these features was found to be 100% accurate in distinguishing aganglionosis from ganglionosis. In the study, it was reported that complete bowel wall thickness did not differ between aganglionic and ganglionic segments or correlate with patient weight or age. The study had been well planned and the results had been comprehensively analysed together with the literature. The manuscript is of acceptable quality.

Sincerely

Round 2

Reviewer 2 Report

Comments and Suggestions for Authors

Congratulations to Fransson et al. on their efforts in revising the manuscript. The authors addressed all of my previous concerns, improving the writing and clarifying some methodological aspects. 

Author Response

Thank you very much for your kind and encouraging feedback. We appreciate your thorough review and are pleased to hear that the revisions have clarified the manuscript and addressed your concerns.

Reviewer 3 Report

Comments and Suggestions for Authors

The article provides an incentive idea and is relevant to the field. However, if this is only the premise for a future technique, this should not be mentioned throughout the article, but in a specific section or paragraph. 

I recommend discussing the UHFUS technique only in the discussion section, as a future promising or future direction technique. Do not state in the conclusion of a possible technique without any specific results.

Author Response

Comment:

The article provides an incentive idea and is relevant to the field. However, if this is only the premise for a future technique, this should not be mentioned throughout the article, but in a specific section or paragraph. 

I recommend discussing the UHFUS technique only in the discussion section, as a future promising or future direction technique. Do not state in the conclusion of a possible technique without any specific results.

Answer: 29/7 2025

Thank you for your thoughtful and constructive feedback.

We appreciate your recommendation and have revised the manuscript accordingly. References to potential future techniques have been limited to the discussion section. We have also removed speculative statements from the conclusion to ensure that only results directly supported by the data are included.

Thank you again for helping us improve the clarity and focus of the manuscript.

Changes made:
Abstract:
Introduction: Intraoperative frozen biopsies are essential during surgery for Hirschsprung’s disease (HD).However, this method has several limitations with the need for a faster and real-time diagnostic alternative. For this, consistent histoanatomical and morphometric differences between aganglionic and ganglionic bowel must be established.
Conclusion (shortened): Histoanatomic layer thickness differs between aganglionic and ganglionic bowel, forming the basis of a diagnostic algorithm. Full bowel wall thickness was independent of patient weight and age.

Introduction:
In this context, further exploration of histomorphometric characteristics may offer valuable insights. Histological differences in bowel wall layer thickness between aganglionic and ganglionic segments have previously been described in fresh ex vivo specimens using manual measurement techniques [6].

Discussion:
Recent studies suggest that tissue differences across various conditions can be identified using ultra-high frequency ultrasound examinations [12, 13]. In the gastrointestinal tract, ultra-high frequency ultrasound has shown potential in distinguishing aganglionic from ganglionic segments by detecting differences in bowel wall architecture [14, 15], and has also been used to identify histomorphological changes associated with bowel inflammation in infants [16].

Conclusion:
The ultimate goal is to translate this knowledge into real-time diagnostic assessment of the bowel wall.

Best regards,
Emma Fransson,

MD PhD student
on behalf of all authors

Round 3

Reviewer 3 Report

Comments and Suggestions for Authors

No additional comments.